# A synthetic planar cell polarity system reveals localized feedback on Fat4-Ds1 complexes

Olga Loza[1†], Idse Heemskerk[2†], Nadav Gordon-Bar[1], Liat Amir-Zilberstein[1], Yunmin Jung[1], David Sprinzak[1]*

[1]Department of Biochemistry and Molecular Biology, Wise Faculty of Life Science, Tel Aviv University, Tel Aviv, Israel; [2]Department of Biosciences, Rice University, Houston, United States

**Abstract** The atypical cadherins Fat and Dachsous (Ds) have been found to underlie planar cell polarity (PCP) in many tissues. Theoretical models suggest that polarity can arise from localized feedbacks on Fat-Ds complexes at the cell boundary. However, there is currently no direct evidence for the existence or mechanism of such feedbacks. To directly test the localized feedback model, we developed a synthetic biology platform based on mammalian cells expressing the human Fat4 and Ds1. We show that Fat4-Ds1 complexes accumulate on cell boundaries in a threshold-like manner and exhibit dramatically slower dynamics than unbound Fat4 and Ds1. This suggests a localized feedback mechanism based on enhanced stability of Fat4-Ds1 complexes. We also show that co-expression of Fat4 and Ds1 in the same cells is sufficient to induce polarization of Fat4-Ds1 complexes. Together, these results provide direct evidence that localized feedbacks on Fat4-Ds1 complexes can give rise to PCP.

DOI: https://doi.org/10.7554/eLife.24820.001

*For correspondence: davidsp@post.tau.ac.il

†These authors contributed equally to this work

Competing interests: The authors declare that no competing interests exist.

## Introduction

Planar cell polarity (PCP) defines the coordinated polarization of cells in the plane of a tissue (*Adler, 2002*; *Lewis and Davies, 2002*; *Lawrence et al., 2007*; *Wang and Nathans, 2007*; *Goodrich and Strutt, 2011*) and underlies the organization and geometry required for the proper function of many developing organs. PCP is usually manifested by orientation of external structures, such as trichomes and sensory hair cells in *Drosophila* (*Goodrich and Strutt, 2011*; *Strutt and Strutt, 2009*), and hair structures in the inner ear and skin of vertebrates (*Montcouquiol et al., 2003*; *Dabdoub and Kelley, 2005*; *Saburi et al., 2008*).

At the molecular level, PCP is defined by asymmetric distribution of transmembrane protein complexes which belong to two families - the Frizzled/Van-Gogh pathway (termed the 'core' pathway) and the Fat/Dachsous (Ft/Ds) pathway. Both were discovered in *Drosophila* but are conserved in higher vertebrates (*Goodrich and Strutt, 2011*; *Singh and Mlodzik, 2012*; *Sharma and McNeill, 2013*).

The main players in the Ft/Ds pathway in *Drosophila* are the large atypical cadherins Ft, Ds and the Golgi protein kinase Four-jointed (Fj). Ft and Ds take part in heterophilic interactions resulting in trans-hetero-complexes on the boundary between cells. Unlike for classical cadherins, there is no evidence of homophilic complexes of either Ft or Ds forming across cells (*Matakatsu and Blair, 2004*; *Matis and Axelrod, 2013*).

The mammalian homologues of Ft and Ds include Fat1-4 and Ds1-2. However, Fat4 and Ds1 have the highest homology to *Drosophila* Ft and Ds, are the most widely expressed, and have the strongest knockout phenotypes (*Rock et al., 2005*). Fat4 and Ds1 null mice show complex morphological

abnormalities in the inner ear, kidney, brain, bone, lymph node, and more. (*Saburi et al., 2008*; *Ishiuchi et al., 2009*). In humans, mutations in Fat4 and Ds1 were recently linked to various cancers and abnormal brain development (*Katoh, 2012*; *Cappello et al., 2013*). Unlike in *Drosophila*, polarized distributions of Fat4 and Ds1 in vertebrates have not been observed yet, probably due to lack of good reagents for staining these proteins in vivo.

It remains poorly understood how polarized distribution of Fat and Ds is established, and how it is coordinated between neighboring cells. Models for the emergence of polarity can be classified according to whether or not they include feedbacks. Models without feedback propose that observed gradients of Ds and Fj are sufficient to explain the asymmetric distribution of Fat-Ds complexes (*Casal et al., 2006*; *Lawrence et al., 2008*; *Strutt, 2009*; *Hale et al., 2015*; *Abley et al., 2013*; *Axelrod and Tomlin, 2011*). However, these models cannot explain how relatively small differences in Ds and Fj expression between neighboring cells can lead to strongly polarized membrane distributions. This problem is solved by models that include localized feedback mechanisms, where we define 'localized' to mean a feedback at the level of protein complexes within the cell boundary, to distinguish it from transcriptional feedback or tissue scale feedbacks. Such models have been previously proposed to explain polarity in the core pathway, where the localized feedbacks are mediated by the cytoplasmic proteins Prickle and Dishevelled (*Amonlirdviman et al., 2005*; *Le Garrec et al., 2006*; *Fischer et al., 2013*). However, no such feedbacks were identified for the Fat-Ds pathway.

In principle, two localized feedback mechanisms can be distinguished: (1) a self-enhancing feedback that promotes the formation of additional Ft-Ds complexes in the same direction of existing Ft-Ds complexes, and (2) a mutual inhibition feedback between complexes in opposite direction (*Figure 1A*). Mathematical modeling predicts that models with both feedbacks can amplify small differences in expression and create strongly polarized cell boundaries (*Mani et al., 2013*; *Burak and Shraiman, 2009*). Intuitively, a model with a self-enhancing feedback alone would lead to roughly equal numbers of opposing complexes, whereas a model with mutual inhibition feedback alone would remove pairs of opposing complexes and leave only the small excess of the dominant complex type. The combination of both feedbacks together leads to strong accumulation of complexes in one direction. Somewhat less intuitive is the prediction of these models that polarity can emerge spontaneously once threshold levels of Ft and Ds are reached, even in the absence of external gradients (*Mani et al., 2013*; *Burak and Shraiman, 2009*). We note that models for PCP based on the core pathway are equivalent to mutual inhibition feedback described above and do not require self-enhancing feedback.

To test whether localized feedbacks are required for establishing Fat4-Ds1 mediated planar cell polarity, it is necessary to directly verify their existence and the prediction of spontaneous polarization in the presence of threshold expression levels. In this work, we develop an experimental synthetic biology platform to study the interactions of Fat4 and Ds1 in live mammalian cells. By analyzing co-cultures of cells expressing Fat4 fused to Citrine and cells expressing Ds1 fused to mCherry, we show that Fat4-Ds1 complex formation exhibits a threshold response to the levels of Fat4 and Ds1, supporting a localized self-enhancing feedback mechanism. We further show that Fat4-Ds1 complexes at the cell boundary form extremely stable clusters, suggesting that complex stabilization through clustering may serve as a mechanism for localized feedback. Finally, we show that cells expressing both Fat4 and Ds1 exhibit strong localized polarization, supporting the existence of mutual inhibition between opposing complexes. We find that the direction of polarization depends on the relative levels of Fat4 and Ds1 across the boundary between cells.

## Results

### Development of an experimental system for quantitative characterization of the mammalian Fat4/Dachsous1 interactions

To study the interactions between Fat4 and Ds1 at the boundary between cells we cloned the full length human Fat4 with a C-terminal fusion of citrine (Fat4-citrine) and the human Ds1 with a C--terminal fusion to mCherry (Ds1-mCherry) (*Figure 1B*). We generated HEK293 stable cell lines expressing Fat4-citrine under a constitutive CMV promoter (HEK-Fat4-citrine) and Ds1-mCherry under a doxycycline inducible promoter (HEK-Ds1-mCherry). Western blotting confirmed that these

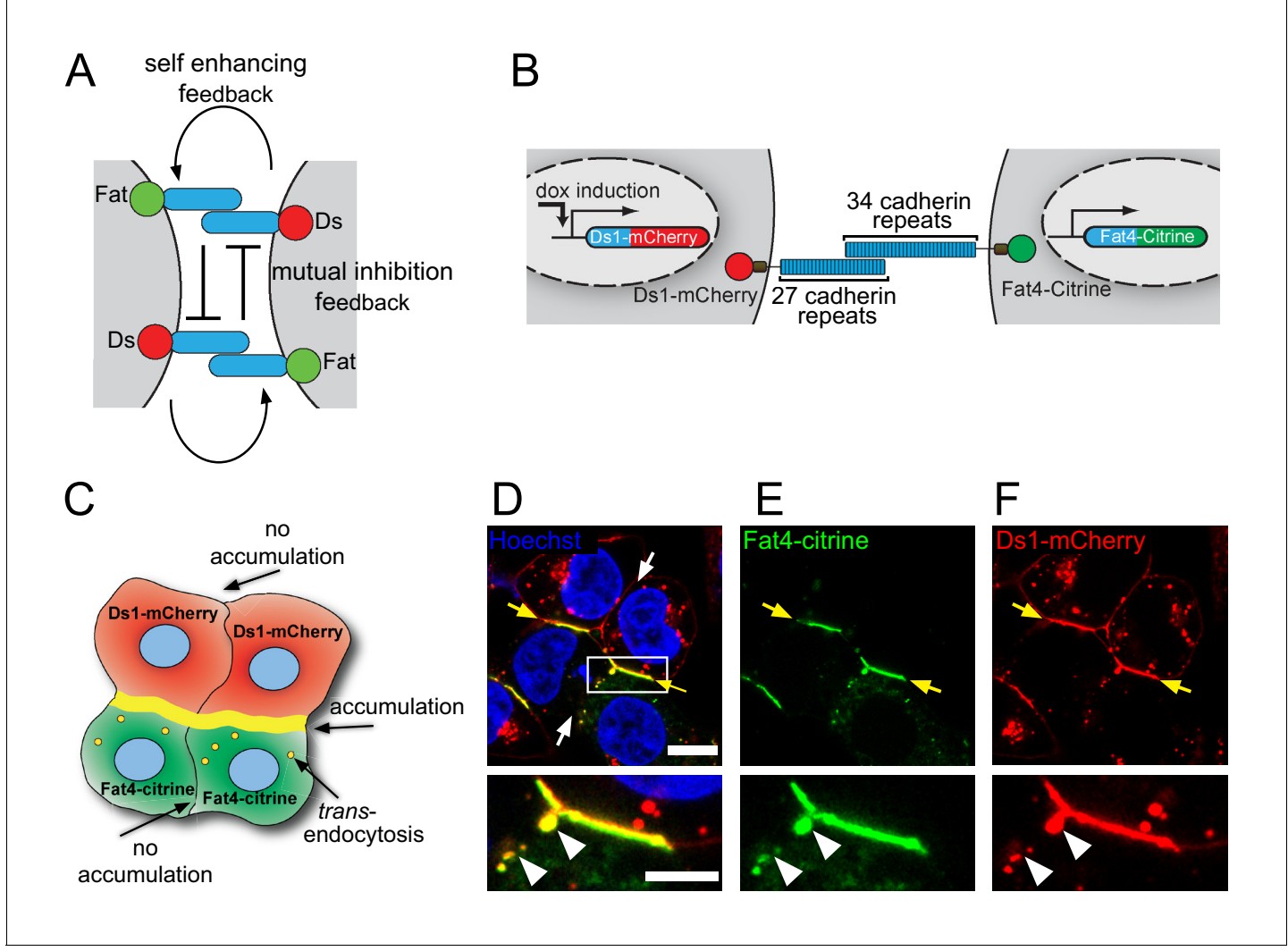

**Figure 1.** Fat4-Citrine and Ds1-mCherry accumulate on heterotypic boundaries. (**A**) Schematic of the localized feedback hypothesis. (**B**) Schematic illustration of the stable cell lines and fusion constructs of Fat4-citrine and inducible Ds1-mCherry. (**C**) Schematic illustration of the cell-cell boundaries formed in a co-culture assay of Fat4-citrine (green) and Ds1-mCherry (red) cells. Yellow boundary represents accumulation at heterotypic boundaries. Yellow vesicles represent trans-endocytosis of Ds1-mCherry into Fat4-citrine expressing cells. (**D–F**) A co-culture of HEK-Fat4-citrine cells (green) and HEK-Ds1-mCherry cells (red). Strong accumulation is observed on heterotypic boundaries (yellow arrows). No accumulation is observed on homotypic boundaries (white arrows). Zoom in on the accumulation area (white box in D) demonstrates that Ds1-mCherry trans-endocytoses (white triangles) into Fat4-citrine expressing cell (but not vice-versa). Nuclei are stained with Hoechst (blue). Scale bar - 20 µm. Supplementary figure (***Figure 1—figure supplement 1***) shows Western blot analysis, monoculture images from Fat4-citrine and Ds1-mCherry cell lines and boundary accumulation in co-culture between MCF7-Fat4-citrine and MCF7-Ds1-mCherry cells.

DOI: https://doi.org/10.7554/eLife.24820.002

The following figure supplement is available for figure 1:

**Figure supplement 1.** Fat4-citrine and Ds1-mCherry accumulate on heterotypic boundaries, but not on homotypic boundaries.

DOI: https://doi.org/10.7554/eLife.24820.003

fusion proteins were expressed and showed the expected molecular bands at ~500 kDa for Fat4-citrine and ~320 kDa for Ds1-mCherry (***Figure 1—figure supplement 1A***).

To study the interactions between Fat4-citrine and Ds1-mCherry we performed co-culture experiments of HEK-Fat4-citrine and HEK-Ds1-mCherry (***Figure 1C–F***). Consistent with the existing evidence for exclusively heterotypic interactions, these co-culture experiments revealed strong accumulation of Fat4-citrine and Ds1-mCherry at heterotypic junctions (yellow arrows in ***Figure 1D–F***), while no accumulation was observed in homotypic junctions (white arrows in ***Figure 1D***) or in

monoculture experiment of either cell line (*Figure 1—figure supplement 1B–C*). Similar accumulation was observed in co-culture experiments of MCF7 cells expressing the same fusion constructs (*Figure 1—figure supplement 1D*).

Surprisingly, we also found that Ds1-mCherry *trans*-endocytoses into the Fat4-citrine cells as evident from numerous vesicles containing both Ds1-mCherry and Fat4-Citrine (white triangles in insets *Figure 1D–F*, see also yellow vesicles in Figures 3 and 4). No trans-endocytosis of Fat4-Citrine into the Ds1-mCherry cells is observed. This observation is the first evidence for the presence of transendocytosis in Fat-Dachsous pathway.

## Accumulation of Fat4-Ds1 complexes requires threshold levels of Ds1 and Fat4

Having confirmed the expected qualitative behavior of our synthetic PCP system, we set out to determine how the accumulation of Fat4-Ds1 complexes depends on the expression levels of Fat4-citrine and Ds1-mCherry. To separately test the effect of self-enhancing and mutual inhibition feedbacks, we first considered co-culture of Fat4-citrine and Ds1-mCherry expressing cells, where mutual inhibition feedback is not present as opposing complexes cannot be formed. We began by analyzing the accumulation of Fat4-citrine and Ds1-mCherry at the population level in snapshots of co-culture experiments with Ds1-mCherry induction periods varying from 0 to 20 hr (*Figure 2A–B*). We used automated image analysis to segment the cells and boundaries, and measure the Fat4-citrine and Ds1-mCherry fluorescence levels in each cell and each boundary (*Figure 2C*). By analyzing tens of thousands of junctions per condition we obtained detailed statistical information about the effect of increased Ds1-mCherry expression on Fat4-Ds1 accumulation.

A self-enhancing positive feedback model is expected to exhibit a non-linear dependence of the accumulation of complexes on the total Ds1-mCherry level in the cell. To test this prediction we measured the fraction of accumulating boundaries and the average Ds1-mCherry expression in images (*Figure 2—figure supplement 1A–B*). We find that relation between these quantities is best fitted with a Hill function with a Hill coefficient of $n = 2.2 \pm 0.3$, whose value $n > 1$ is consistent with a localized positive feedback model (*Figure 2D*). A somewhat stronger non-linear response ($n = 4.5 \pm 1.3$) was observed in a second experiment where Ds1 levels where induced to lower levels (*Figure 2—figure supplement 2A–C*).

Having analyzed the population averages we next wanted to test if accumulation on a cell boundary is associated with locally higher expression levels of Fat4-citrine and Ds1-mCherry. For each Ds1 induction time, we compared the distributions of Fat4 and Ds1 expression in cells that have strong accumulation on their boundary ('accumulating boundaries', *Figure 2E–F*, dashed lines) and cells that do not ('non-accumulating boundaries', *Figure 2E–F*, solid lines). We found that cells with accumulating boundaries generally express higher levels of either Fat4 or Ds1 (*Figure 2E–F*, solid vs. dashed lines, 20 hr time point). We note that even with long induction times not all Fat4-Ds1 boundaries exhibit accumulation.

To test if accumulation on boundaries simultaneously requires high levels of Fat4-citrine and high levels of Ds1-mCherry, we plotted for each induction time, the two-dimensional distribution of the levels of Fat4-citrine and Ds1-mCherry in the cells flanking all boundaries (*Figure 2G–H*, *Figure 2—figure supplement 1C*, *Figure 2—figure supplement 2C*) (In *Figure 2H*, each dot represents one boundary and the axes values indicate the levels of Fat4-Citrine and Ds1-mCherry in the cells flanking that boundary). We observed a clear separation between 'accumulating boundaries' (yellow dots in *Figure 2H*) and 'non-accumulating boundaries' (boundaries without accumulation) (purple dots in *Figure 2H*) indicating that accumulation occurs when both Fat4-citrine and Ds1-mCherry are expressed above a certain threshold level. Note that, while the fraction of boundaries exhibiting accumulation increased with induction time (*Figure 2—figure supplement 1C*, *Figure 2—figure supplement 2D*), the separation between the two populations (i.e the threshold levels) remained almost the same (dashed lines *Figure 2H*). Such a behavior is expected from a localized feedback model exhibiting bistability.

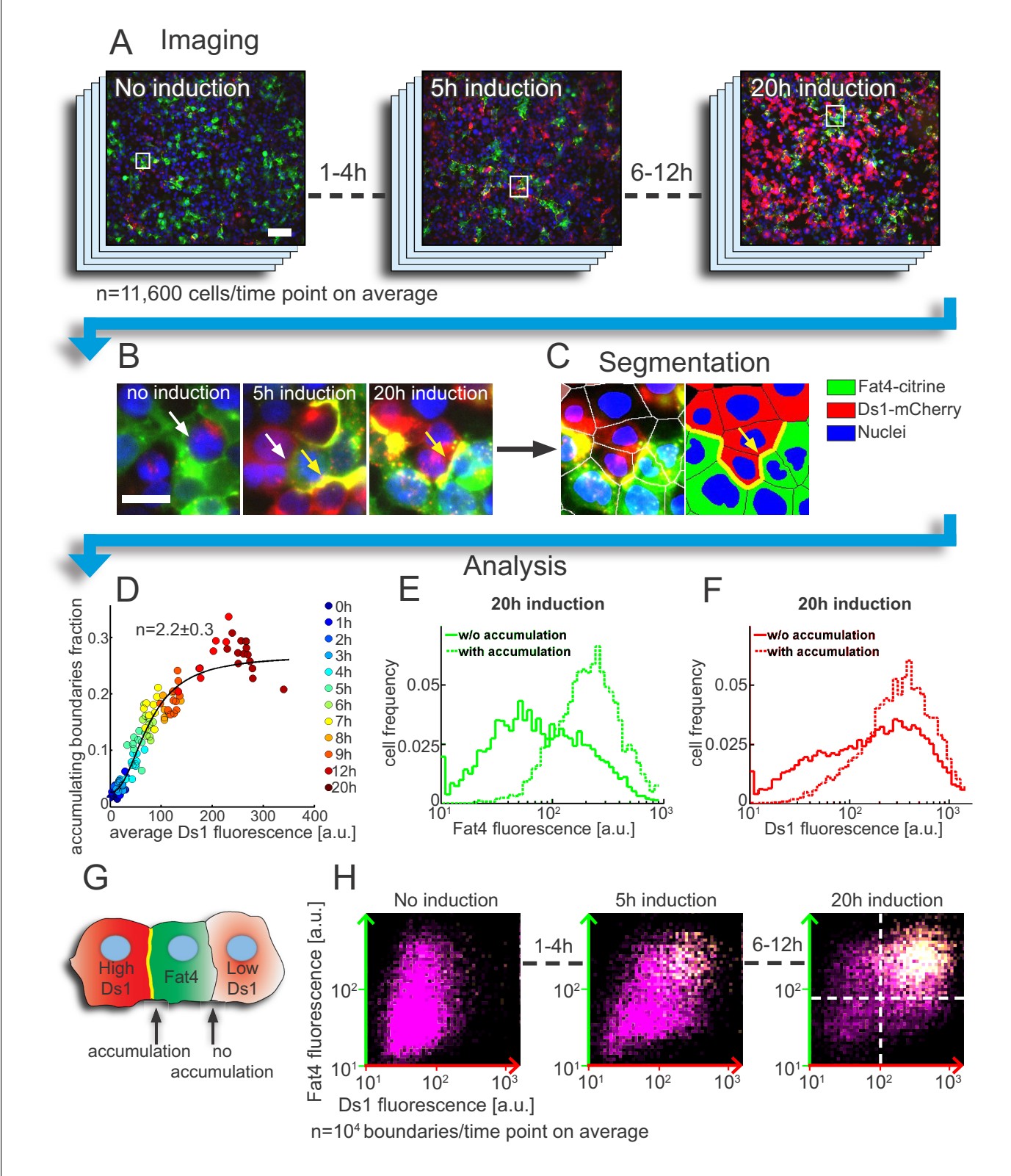

**Figure 2.** Accumulation on the boundary between cells requires threshold levels of Fat4-citrine and Ds1-mCherry. (A–C) The analysis pipeline for Fat4-Ds1 boundary accumulation. (A) Snapshots of HEK-Fat4-citrine (green) and HEK-Ds1-mCherry (red) co culture, at different Ds1 induction times. Nuclei are stained with Hoechst (blue). Higher Ds1-mCherry (red) levels are observed for longer induction times. Scale bar −100 μm. (B) Zoom in on the areas marked with rectangles in (A) show both accumulating (yellow arrows) and non accumulating (white arrows) boundaries. Scale bar – 20 μm. (C)

*Figure 2 continued on next page*

*Figure 2 continued*

Segmentation of the 20 hr induction time point (right image in (B)). Left image shows overlay of the cell segmentation while right image shows the segmentation label for cell type and boundary accumulation (green – Fat4, red – Ds1, yellow – accumulating boundary, blue – nuclei). (D) Plot showing the increase in the fraction of accumulating boundaries with Ds1-mCherry levels. Different colors represent different doxycycline induction times. Hill function fit (solid line) gives a Hill coefficient of $n = 2.2 \pm 0.3$, showing nonlinear increase. The error on $n$ represents 95% confidence interval of the fit. (E–F) Probability distribution functions (pdf) of the total (cytoplasm +boundary) Fat4-citrine levels (E) and Ds1-mCherry levels (F) in cells exhibiting accumulation on heterotypic boundaries (dashed lines) and in cells not exhibiting accumulation on heterotypic boundaries (solid lines). Pdf's shown are for the case of 20 hr doxycycline induction time. (G) Schematic of the defined 'accumulating' and 'non-accumulating' boundaries. (H) Two dimensional distributions of the expression levels of Fat4-citrine and Ds1-mCherry in cells flanking each boundary after 0, 5 and 20 hr induction with doxycycline. The brightness in the distribution corresponds to the frequency with which given levels of Ds1-mCherry (x-axis) and Fat4-citrine (y-axis) flank Fat4-Ds1 boundaries (see schematic in G). Both axes are on a logarithmic scale. The clear separation between 'accumulating boundaries' (yellow) and 'non-accumulating boundaries' (purple) indicates the threshold concentrations of Ds1 and Fat4 (dashed lines) above which a boundary is formed. Supplementary figure 1 (*Figure 2—figure supplement 1*) shows the average Ds1-mCherry expression, fraction of accumulation, and the distributions of accumulating and non-accumulating boundaries at all induction times. Supplementary figure 2 (*Figure 2—figure supplement 2*) shows the results of a duplicate experiment but with slightly different Ds1 induction rates.

DOI: https://doi.org/10.7554/eLife.24820.004

The following source data and figure supplements are available for figure 2:

**Source data 1.** A source data used to produce *Figure 2*, *Figure 2—figure supplement 1* and *Figure 2—figure supplement 2*.
DOI: https://doi.org/10.7554/eLife.24820.007
**Figure supplement 1.** Accumulation on the boundary requires high levels of both Fat4 and Ds1.
DOI: https://doi.org/10.7554/eLife.24820.005
**Figure supplement 2.** Accumulation on the boundary requires high levels of both Fat4 and Ds1.
DOI: https://doi.org/10.7554/eLife.24820.006

## Live imaging reveals threshold response of Fat4-Ds1 accumulation dynamics at the single cell level

Analysis of the snapshots demonstrated a threshold dependence of Fat4-Ds1 complexes on Fat4-citrine and Ds1-mCherry levels, however it does not show the time scale of accumulation on individual boundaries. Moreover, the non-linear response in population level measurements may be smeared by averaging over many cells. To test whether a threshold response is also observed at the single cell level, and how abrupt the onset of accumulation is, we performed live cell imaging of single Fat4-Ds1 boundaries during Ds1-mCherry induction.

To look at the accumulation of single boundaries over time we performed experiments using the two-cell assay – a micropatterning based method previously used by us and others to look at single pairs of cells over extended time periods (*Desai et al., 2009*) (*Shaya et al., 2017*) (*Figure 3A*). We also tracked single Fat4- and Ds1-expressing cell pairs in a free co-culture assay (*Figure 3—figure supplement 1A*). We then quantified the total Fat4 and Ds1 levels as well as the accumulation of Fat4-Ds1 complexes on the cell boundary over time after induction of Ds1-mCherry expression (*Figure 3B* and *Video 1*, *Figure 3—figure supplement 1A–C*). While the total Fat4 levels do not change, the total Ds1 levels increase slowly above background level, and become observable about 100 min after addition of doxycycline (see *Figure 3—figure supplement 2A–C*). This delay in observed fluorescence is probably due to maturation time of mCherry, as mRNA levels increase linearly upon induction (*Figure 3—figure supplement 2D*). We note that most Ds1-mCherry is localized on the cell membrane, even in the absence of Fat4 expressing cells (*Figure 3—figure supplement 2A–C*), and that the measured total and membrane Ds1 levels are proportional, justifying the use of total Ds1-mCherry as a measure of the relevant Ds1 level in the cell.

The accumulation of Fat4-citrine and Ds1-mCherry proteins on the boundary between cells shows a sharp increase shortly after Ds1 levels starts increasing (*Figure 3C*, *Figure 3—figure supplement 1A–C*). Furthermore, Ds1-mCherry on the boundary continues to increase faster than the total Ds1 (i.e. in a non-linear fashion) for several hours after accumulation starts (*Figure 3—figure supplement 1F*), consistent with a threshold response. Taken together, the population snapshot experiments (*Figure 2*) and the single cell time lapse movies (*Figure 3*) strongly support a non-linear threshold response of Fat4-Ds1 complexes to the levels of Fat4 and Ds1.

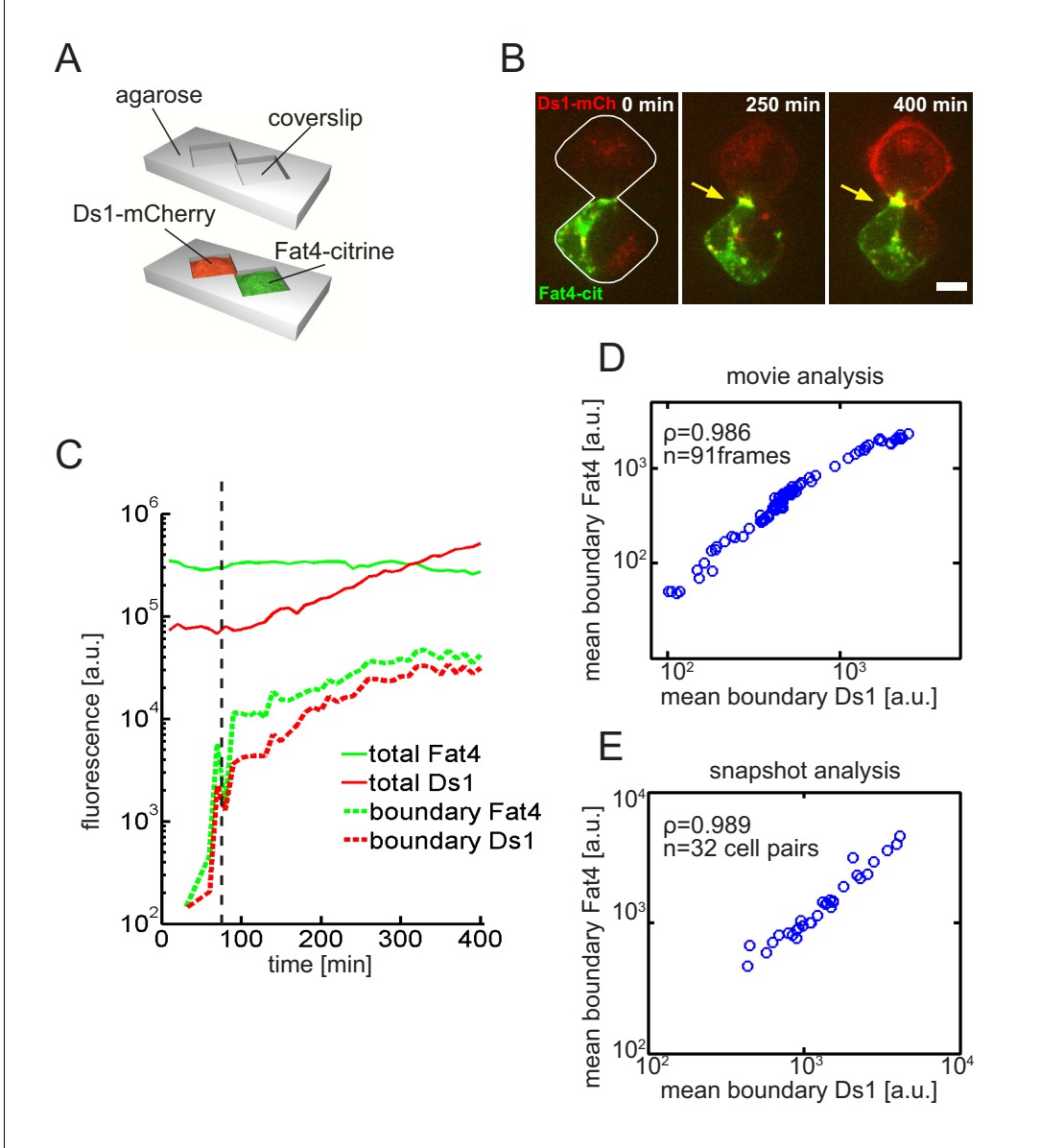

**Figure 3.** Live imaging of Fat4-Ds1 accumulation dynamics reveals threshold response to Ds1 levels at the single cell. (A) Schematic of the two-cell assay. In this assay two cells are restricted to a bowtie-shaped microwell allowing imaging of accumulation dynamics over time. (B) A filmstrip showing a movie in the two-cell assay with HEK-Fat4-citrine cell (green) co-cultured with a HEK-Ds1-mCherry cell (red) (see *Video 1*). Imaging started after the addition of the 100 ng/ml doxycycline. Each image in the filmstrip is a sum of 8 z-slices encompassing the total width of the cells. As Ds1 levels increases, both proteins co-localize and accumulate at the cell boundary (yellow arrow). Scale bar - 10 µm. (C) Quantitative analysis of accumulation dynamics. The levels of total cellular Fat4-citrine (green solid line), total cellular Ds1-mCherry (red solid line), boundary Ds1-mCherry (red dashed line), and boundary Fat4-citrine (green dashed line) are plotted as a function of post-induction time. The fluorescence of both proteins exhibit a threshold response (black dashed line). (D–E) Mean boundary levels of Fat4-citrine and Ds1-mcherry are proportional to each other. Analysis of the single cell movie (D) and snapshots (E) shows that Fat4 and Ds1 fluorescence at the accumulating boundary are proportional to each other. ρ and n, correspond to the Pearson correlation coefficient and the number of frames, respectively. Supplementary figure 1 (*Figure 3—figure supplement 1*) shows accumulation dynamics of free co-culture experiments and the non-linear accumulation of all movies shown here. Supplementary figure 2 (*Figure 3—figure supplement 2*) shows the distribution and dynamics of membrane Ds1 vs. total Ds1 in the cell.

DOI: https://doi.org/10.7554/eLife.24820.009

The following figure supplements are available for figure 3:

**Figure supplement 1.** Live imaging of Fat4-Ds1 accumulation dynamics in free co-culture reveals threshold response to Ds1 levels.
DOI: https://doi.org/10.7554/eLife.24820.010

**Figure supplement 2.** The membrane fraction of Ds1 mCherry is large and proportional to the total Ds1-mCherry in the cell.

*Figure 3 continued on next page*

*Figure 3 continued*

DOI: https://doi.org/10.7554/eLife.24820.011

## Fat4 and Ds1 accumulation are proportional, consistent with stoichiometric binding

Although Fat4 and Ds1 accumulate on the boundary between cells, it is not clear whether this accumulation reflects the formation of heterotypic complexes or the independent recruitment of unbound Fat4 and Ds1 on both sides of the boundary. To distinguish between these two scenarios we analyzed the relative accumulation of both proteins in single cell movies. It is expected that feedback on complex formation would maintain a fixed stoichiometric ratio between Fat4 and Ds1, while independent accumulation (or independent feedback) of unbound Fat4 and Ds1 would not. Analysis of the three movies described above (*Figure 3C* and *Figure 3—figure supplement 1B–C*) showed that Fat4 and Ds1 fluorescence at the accumulating boundary are proportional as the boundary is formed (spanning over two orders of magnitude in fluorescence) (*Figure 3D* and *Figure 3—figure supplement 1D–E*). Similar results were obtained by analyzing snapshots from 31 boundaries showing, a high degree of correlation ($\rho = 0.989$) over two orders of magnitudes in fluorescence (*Figure 3E*). This linear relation between the accumulation on both sides of the boundary indicates the formation of Fat4-Ds1 complexes with a fixed stoichiometric ratio.

## Fat4 and Ds1 form extremely stable complexes on the boundary

We next wanted to understand the mechanism of localized self-enhancing feedback. It has been previously shown that cadherins can interact cooperatively across the cell boundary to promote cell adhesion (*Chen et al., 2005*; *Zhang et al., 2009*). We therefore hypothesized that stabilization of Fat4-Ds1 complexes, possibly through clustering, can be an underlying mechanism for the self-enhancing feedback. Although strictly, such a mechanism would predict increased complex stability at increased complex concentration, one would at the very least expect a large difference in stability between bound Fat4-Ds1 complexes and unbound Fat4 and Ds1. Consistent with this idea, increased stability of Ft-Ds complexes was recently observed in localized puncta on cell boundaries in *Drosophila* (*Hale et al., 2015*).

To test the stability of Fat4-Ds1 complexes, we performed fluorescence recovery after photobleaching (FRAP) experiments on boundaries exhibiting accumulation (*Figure 4A–B*, *Video 2*). Experiments were performed by bleaching the Fat4-citrine fluorescence (green in *Figure 4A*), without affecting the Ds1-mCherry fluorescence (red in *Figure 4A*), allowing automated tracking of the recovery dynamics of Fat4-citrine on the boundary. Analysis of the recovery profile over time showed very slow recovery time (>25 min) on the accumulating boundary (see boundary on kymograph in *Figure 4B*), reflecting the extremely low turnover and membrane dynamics of Fat4-Ds1 complexes.

We then examined whether the slow recovery of complexes is a reflection of the slow dynamics of unbound Fat4 and Ds1 or of the increased stability of the complex. To analyze the membrane dynamics of Fat4 and Ds1 we used FRAP combined with total internal reflection fluorescence microscopy (FRAP-TIRF) (*Khait et al., 2016*). We performed these experiments on monocultures of Fat4-citrine (*Figure 4C–D*, *Video 3*) and Ds1-mCherry (*Figure 4—figure supplement 1A–B*). Both Fat4-citrine and Ds1-

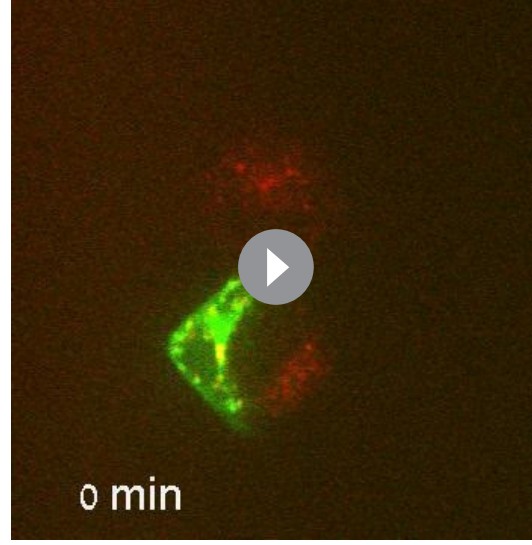

**Video 1.** A timelapse movie showing the dynamics of Fat4-Ds1 accumulation in a single cell pair. Movie used to generate filmstrip in *Figure 3B*.

DOI: https://doi.org/10.7554/eLife.24820.008

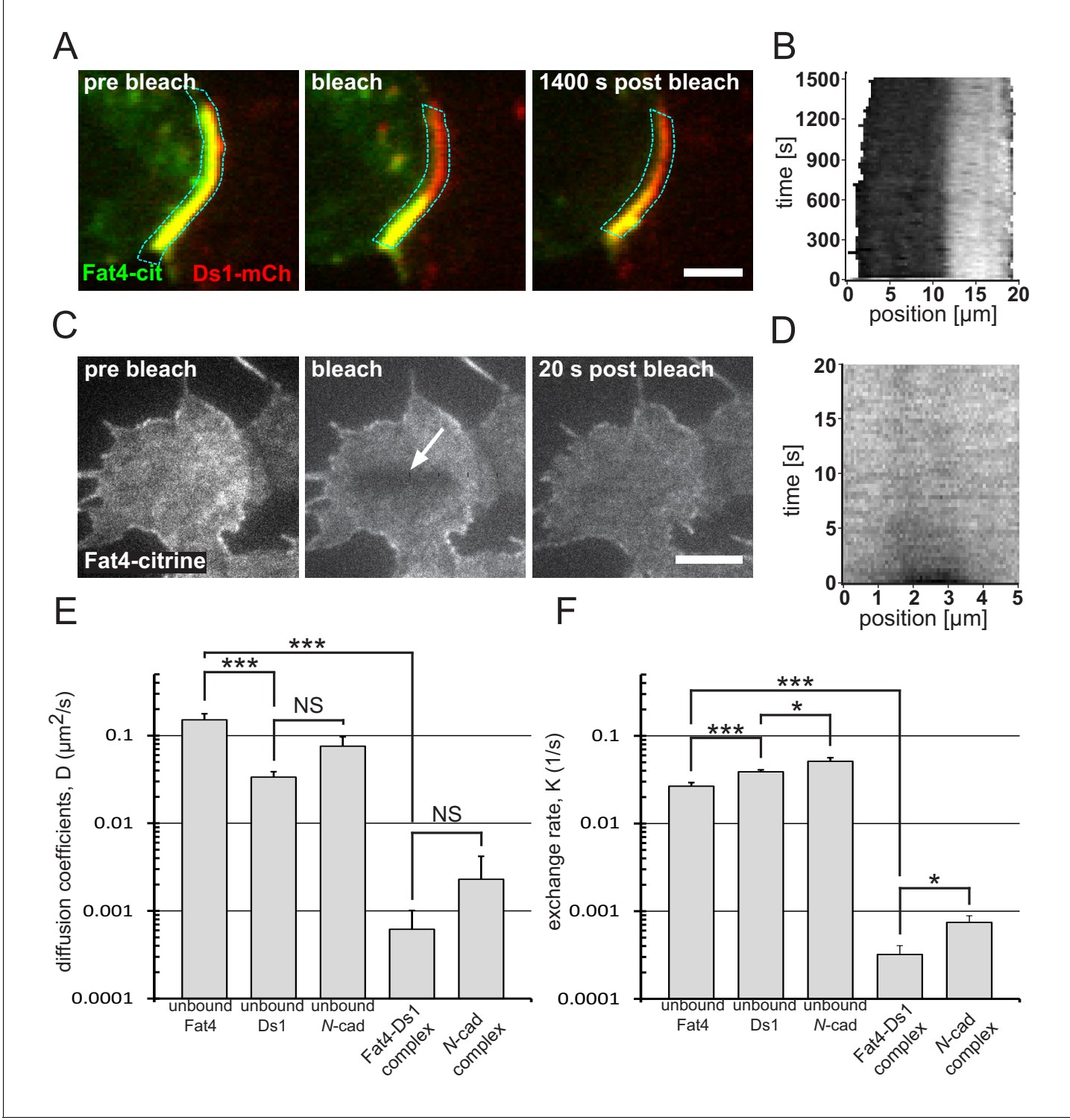

**Figure 4.** Bound Fat4-Ds1 complexes on the boundary are more stable than unbound Fat4 and Ds1. (A) A filmstrip showing a fluorescence recovery after photobleaching (FRAP) experiment on a boundary exhibiting accumulation (yellow) of Fat4-citrine (green) and Ds1-mCherry (red). (see *Video 2*) (B) A kymograph showing the fluorescence recovery profile along the boundary outlined in blue in (A). The fluorescence level (gray scale) is shown as a function of the position along the boundary (x-axis), and the time after photobleaching (y-axis). (C) A filmstrip from FRAP-TIRF experiment on a cell that express Fat4-citrine (see *Video 3*). Arrow indicates the bleached area. (D) A kymograph showing the fluorescence recovery profile in the bleached area in (C). Almost full recovery of the bleached area is obtained after 20 s. Scale bars - 5 μm. (E–F) Distributions of Diffusion coefficients (E) and exchange rates (F) obtained from analysis of FRAP experiments as those shown in (A–D). * and *** correspond to p-value<0.05 and p-value<0.001, respectively, as

*Figure 4 continued on next page*

*Figure 4 continued*

estimated by t-test. The number of experiments for each sample are: unbound Fat4 n = 29, unbound Ds1 n = 36, unbound *N*-cadherin n = 21, Fat4-Ds1 complex n = 10, *N*-cadherin complex n = 11. Error bars correspond to SEM. Supplementary figure (*Figure 4—figure supplement 1*) shows the analysis for unbound Ds1-mcherry, unbound *N*-cadherin-GFP, and the bound *N*-cadherin complex.

DOI: https://doi.org/10.7554/eLife.24820.013

The following figure supplement is available for figure 4:

**Figure supplement 1.** Unbound Ds1-mCherry and *N*-cadherin-GFP exhibit fast membrane dynamics.

DOI: https://doi.org/10.7554/eLife.24820.014

mCherry exhibited very fast recovery compared to the Fat4-Ds1 complexes, of the order of a few seconds (*Figure 4B and D* and *Figure 4—figure supplement 1B*). Quantitative analysis of FRAP movies allowed estimating the effective diffusion coefficients and membrane-cytoplasm exchange rates of Fat4-Ds1 complexes as well as unbound Fat4 and Ds1 (see Materials and methods). We found that the mean diffusion coefficient of Fat4-Ds1 complexes is more than two orders of magnitude smaller than that of unbound Fat4 and Ds1 (*Figure 4E*). Similarly, the mean exchange rate of Fat4-Ds1 complexes are significantly smaller than those of the unbound Fat4 and Ds1 (*Figure 4F*). Hence, complexes formed on the boundary between cells show extremely high stability compared to unbound Fat4 and Ds1.

To test if the observed slow dynamics is specific to Fat4-Ds1 complexes we also analyzed the dynamics of both unbound and bound *N*-cadherin-GFP (*N*-cad-GFP) (*Figure 4—figure supplement 1C–F*) using the same experimental approach. Similar to the results with Fat4 and Ds1, we found that accumulation of *N*-cad-GFP complexes on the cell boundary exhibited significantly slower dynamics than unbound *N*-cad-GFP (*Figure 4E–F* and *Figure 4—figure supplement 1C–F*). This result is consistent with cooperative binding and enhanced stabilization of complexes at the boundary as previously observed for E-cadherin (*de Beco et al., 2009*). Overall, these results support a model where stabilization of Fat4-Ds1 complexes via clustering serves as a mechanism for localized self-enhancing feedback.

Finally, we also note that unbound Fat4 exhibited significantly faster membrane diffusion than unbound Ds1 (*Figure 4E*) despite having a significantly higher molecular weight (500 KDa vs 320 KDa). This suggests that either Ds1 diffusion is somehow inhibited compared to Fat4, or that the surface dynamics of Fat4 are enhanced through active transport processes.

## Boundary accumulation of Fat4 and Ds1 exhibits a 100–200 nm gap

Previous work on *Drosophila* Ft and Ds showed that Ft-Ds complexes form puncta along the cell boundaries (*Hale et al., 2015*; *Brittle et al., 2010*). To examine the spatial structure of Fat4-Ds1 complexes in our system we performed high resolution imaging of these boundaries. Unlike the *Drosophila* case, but consistent with a previous report (*Ishiuchi et al., 2009*) in mammalian cells, we mostly observe continuous accumulation along the boundary. However, surprisingly, we observed a spatial gap between the peak in red fluorescence (Ds1-mCherry) and the peak in green fluorescence (Fat4-citrine) along the boundary (*Figure 5A–C*). We refer to this gap as the 'rainbow' feature (*Figure 5B*). Rainbows were apparent in most boundaries exhibiting accumulation in both HEK293 and MCF7 cells (*Figure 5—figure supplement 1A–H*). This shift is not due to chromatic aberration,

as images of cells taken with fluorescent beads show clear rainbows even after the chromatic aberration was corrected (*Figure 5—figure supplement 1I–N*). Furthermore, we observed that different boundaries in the same picture and even the same cell (when both Fat4 and Ds1 are co-expressed) can exhibit oppositely directed rainbows (see *Figure 6B*). A similar shift was observed in images taken in super resolution STED microscopy (*Figure 5—figure supplement 1E–H*).

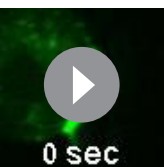

**Video 2.** A timelapse FRAP movie showing the dynamics of bound Fat4 in Fat4-Ds1 complex on the accumulating boundary. Movie used to generate filmstrip in *Figure 4A*.

DOI: https://doi.org/10.7554/eLife.24820.012

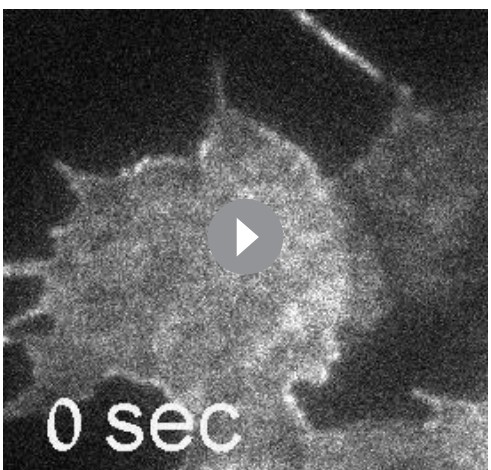

**Video 3.** A timelapse FRAP-TIRF movie showing the dynamics of the unbound Fat4 on the basal membrane of the Fat4 expressing cell. Movie used to generate filmstrip in *Figure 4C*. Github repository - All analysis code can be found at *Loza, 2017*.
DOI: https://doi.org/10.7554/eLife.24820.015

Quantitative analysis of 61 boundaries revealed a relatively tight distribution of gaps with mean and standard deviation of 116 ± 63 nm (*Figure 5D*). Such a gap between the c-termini of Fat4 and Ds1 (where the fluorophores are fused to) is consistent with the length of Fat4 and Ds1 bound to each other in an extended linear form. Given that the length of a cadherin ectodomain is 4.5 nm (*Leckband and Prakasam, 2016*), we estimate the extended form of the complex should be at least 150 nm ($34 \times 4.5nm = 153nm$ assuming full overlap between Fat4 and Ds1) (*Figure 1B*).

## Cells expressing both Fat4 and Ds1 exhibit localized polarization

Polarity in vivo is established in cells expressing both Fat and Ds, with Ds and Fat activity or expression controlled by a morphogen gradient (*Hale et al., 2015*; *Rogulja and Irvine, 2005*; *Rogulja et al., 2008*). The localized feedback model, however, predicts that polarization can emerge spontaneously; i.e. without external gradients (*Mani et al., 2013*). To test if our synthetic cell culture system can polarize without tissue scale gradients we generated stable HEK293 cells that expressed both Fat4-citrine and inducible Ds1-mCherry (*Figure 6A*). Upon induction of Ds1-mCherry expression we observed clear accumulation of Fat4-citrine and Ds1-mCherry on the boundary between cells (*Figure 6B*), similar to the accumulation observed in co-culture of Fat4-citrine and Ds1-mCherry cells (*Figure 1D–F*).

Analysis of high resolution images (*Figure 6C*) revealed rainbows similar to the ones observed in co-culture experiments (*Figure 5*). The observed rainbows show that even in cells expressing both Fat4-citrine and Ds1-mCherry, there is a strong bias in the direction of complex formation on any given cell-cell boundary. Furthermore, we generally do not see domains of opposite polarity within single boundaries suggesting that complexes on cell boundaries align in a coordinated manner. This observation is consistent with mutual inhibition feedback between complexes and suggests that direct interaction between opposing complexes is sufficient for generating polarity.

## The direction of polarity depends on differences in Fat4 and Ds1 expression

Interestingly, we see that the direction of polarity, i.e., the direction from green to red in the rainbow (black arrow in *Figure 6D*), can differ between boundaries of the same cell (see, for example, boundaries 2, 3, and 4, in *Figure 6B–D*). This behavior is different from the long range polarization of Ft-Ds observed in *Drosophila* wing and larvae (*Lawrence et al., 2008*; *Brittle et al., 2010*; *Rogulja et al., 2008*; *Aigouy et al., 2010*), but is reminiscent of polarity reversal within the same cell recently observed in *Drosophila* larvae denticles (*Rovira et al., 2015*).

We hypothesized that the difference between the localized polarity observed in the cell culture system and the coordinated polarity observed in vivo may be due to the variability in expression of Fat4 and Ds1 in our system and that this variability results in expression differences across the boundaries that locally bias the polarity. To check this hypothesis, we measured the differences in cytoplasmic levels of Fat4 and Ds1 across each boundary. We defined the direction of expression gradients as the directions going from low to high expression level of Fat4 and Ds1 (marked by red and green triangles in *Figure 6D*). We then checked whether the direction of polarization, determined by the rainbow feature (red-green polarity bar in *Figure 6D*), is aligned with the direction of expression gradients of Fat4 and Ds1 across the boundary. We found that when the expression gradients of Fat4 and Ds1 are opposed (such as in boundaries 1,2, and 4 in *Figure 6D*) the polarity aligns with both the Ds1 and Fat4 gradients. Namely, that boundary accumulation of Ds1 occurs on

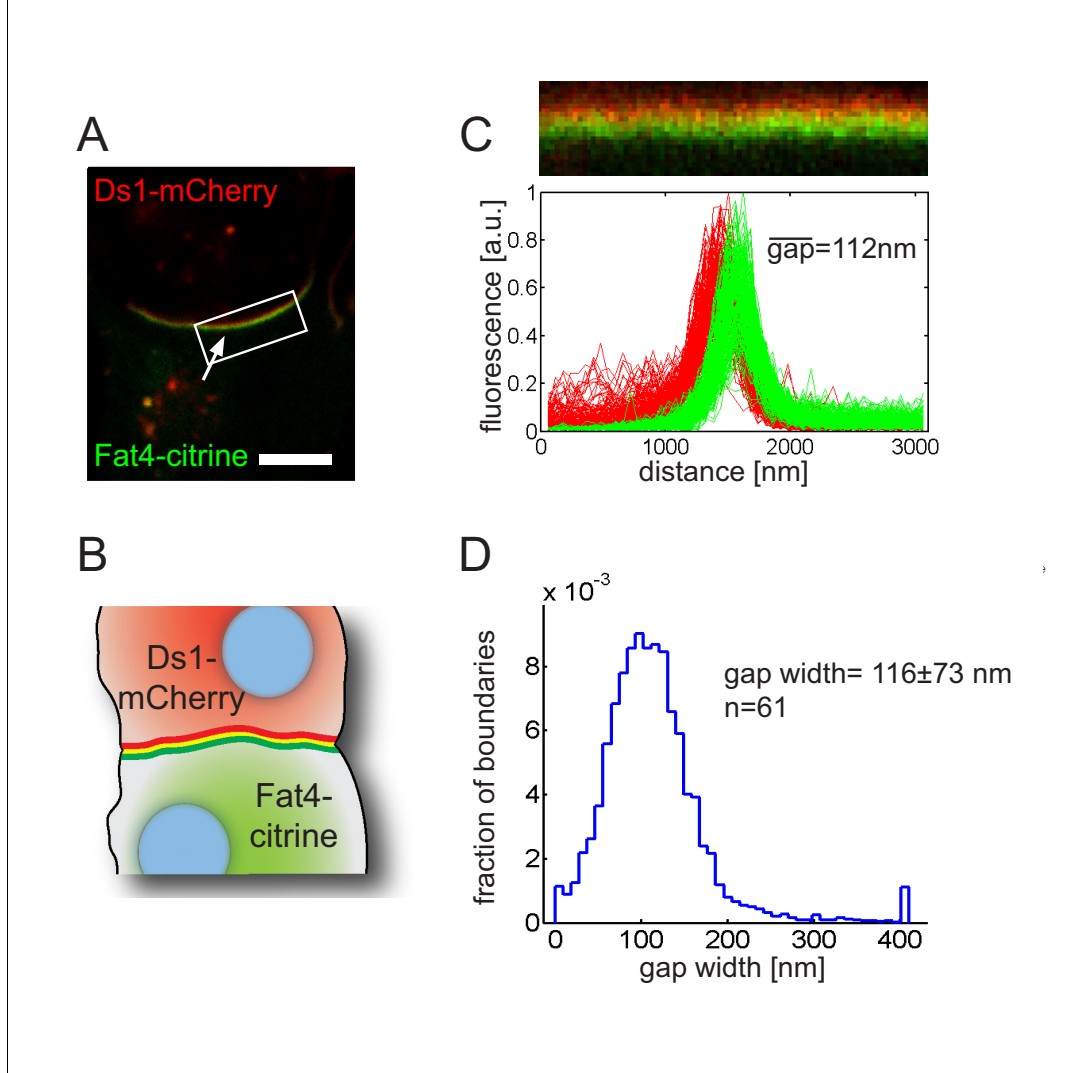

**Figure 5.** Fat4-citrine and Ds1-mCherry fluorescence at the boundary between cells are shifted by 100–200 nm. (**A**) A high resolution image of a boundary exhibiting a 'rainbow' feature (composed of three stripes green, yellow and red; white arrow) indicating a shift between red and green fluorescence. Scale bar - 5 µm. (**B**) An illustration of the observed 'rainbow' feature. (**C**) A straightened version of the boundary shown in A (top). Fluorescence profiles (bottom) of Fat4-citrine (green) and Ds1-mCherry (red) along lines perpendicular to the boundary. Mean gap size for this boundary is as indicated (**D**) Probability distribution function of the distance between the peaks in the fluorescence profiles. Mean gap width for 61 boundaries as indicated. Supplementary figure (*Figure 5—figure supplement 1*) shows control experiments in MCF7 cells, super resolution STED images, and rainbows after correction of chromatic aberrations.

DOI: https://doi.org/10.7554/eLife.24820.016

The following figure supplement is available for figure 5:

**Figure supplement 1.** The rainbow feature is observed in other cell types and with super resolution microscopy.

DOI: https://doi.org/10.7554/eLife.24820.017

the side of the cell with higher Ds1 expression, and conversely, boundary accumulation of Fat4 occurs on the side of the cell with higher Fat4 expression. However, in situations where both expression gradients are aligned (such as in boundary 3 in *Figure 6D*) we found that the direction of polarity can align either with the Fat4 gradient or with the Ds1 gradient. Repeating this analysis on 107 boundaries (*Figure 6E*) indeed showed that if Fat4 and Ds1 expression gradients are opposed, then the polarity almost always aligned with the direction of both gradients (bottom pie chart in *Figure 6E*). On the other hand, when Ds1 and Fat4 expression gradients were aligned (top pie chart in *Figure 6E*) we got almost equal number of boundaries aligned with Ds1 gradient (33 out of 68) and boundaries aligned with the Fat4 gradient (27 out 68). This result is consistent with a situation

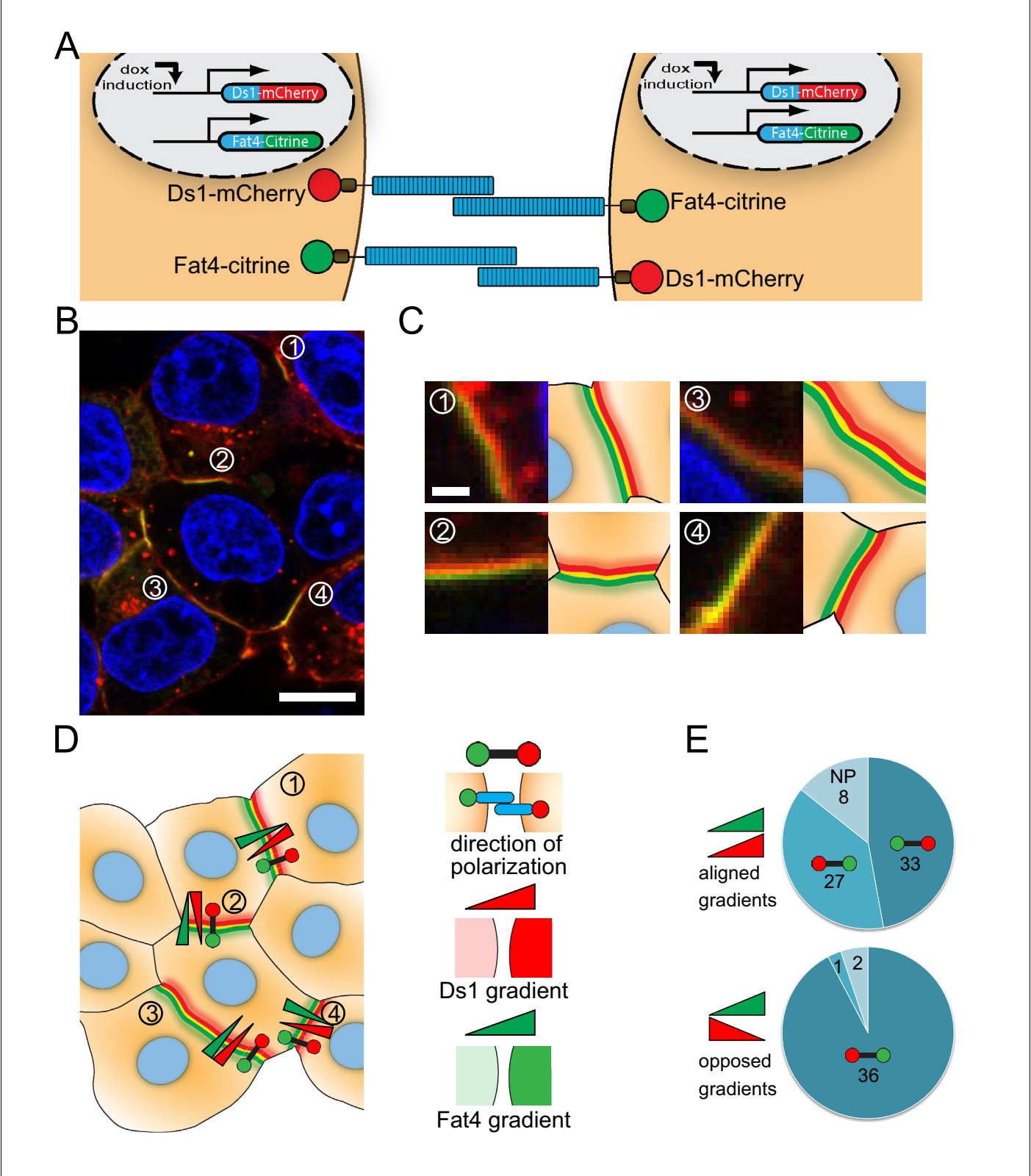

**Figure 6.** Fat4-citrine and Ds1-mCherry polarize in cells expressing both proteins. (**A**) Schematic illustration of the stable cell lines expressing both Fat4-citrine and inducible Ds1-mCherry in the same cell. (**B**) An image of HEK293 cells expressing both Fat4-citrine and Ds1-mCherry. A rainbow feature (composed of three stripes green, yellow and red) is evident at the boundary between the cells. Scale bar - 10 μm. (**C**) Zoom in on the boundaries in (**B**) marked by the numbers 1–4. Each boundary is presented next to its schematic illustration. Scale bar - 1 μm (**D**) An illustration of all the cells and

*Figure 6 continued on next page*

*Figure 6 continued*

boundaries shown in (B). The red-green barbells indicate the direction of polarity as determined by analysis of the rainbow. In this notation, the red and green circles marks the 'red side' and the 'green side' of the rainbow, respectively (see schematic of the notation on the right panel). The red and green triangles represent the directions of the cytoplasmic Ds1 and Fat4 gradients between the two cells flanking the boundary, respectively (cytoplasmic levels where measured in the area adjacent to the boundary – see Materials and methods). (E) Pie charts showing how the direction of polarization (red-green barbell) aligns with either the Fat4 expression gradient (green triangle), or the Ds1 expression gradient (red triangle), or both, in the 107 analyzed boundaries. In the boundaries where the Fat4 and Ds1 gradients are opposed (bottom pie chart) the polarity almost always (36 out of 39) aligns in a direction compatible with both gradients. In the boundaries where the Fat4 and Ds1 gradients are aligned (top chart), the polarity cannot be compatible with both gradients. In these cases, it aligns with the Fat4 gradient in about half of the boundaries (27 out of 68), and with the Ds1 gradient in the other half (33 out of 68). NP – Non-polarized boundaries (no clear rainbow observed). Supplementary figure (*Figure 6—figure supplement 1*) shows that the polarization aligns with the expression gradient that existed prior to boundary accumulation.

DOI: https://doi.org/10.7554/eLife.24820.018

The following figure supplement is available for figure 6:

**Figure supplement 1.** Polarization aligns with the expression gradient that existed prior to boundary accumulation.
DOI: https://doi.org/10.7554/eLife.24820.019

---

where the polarity is either controlled by the Ds1 expression gradient or controlled by the Fat4 expression gradient, depending which of the two is dominant.

To check whether the final direction of polarity reflects the expression gradients that existed prior to the formation of polarized accumulation, we performed live imaging of boundary accumulation using confocal Airyscan technology. The filmstrip from such a movie (*Figure 6—figure supplement 1*) indeed showed that the rainbow feature emerged in the direction of the Ds1 expression gradient that existed before bundary accumulation occured (no significant Fat4 expression gradient is observed in this movie). Overall, these observations support a picture where localized polarization of Fat4-Ds1 complexes in cell culture is biased by local gradients generated by cell-to-cell variability.

## Discussion

In this work, we have adopted a synthetic biology approach to planar cell polarity and reconstituted Fat4-Ds1 PCP in a cell culture system to elucidate the basic mechanisms underlying polarity establishment. Reducing the in vivo system to its core allows dissecting Fat4-Ds1 interactions in a controlled and quantitative manner, beyond what is possible in vivo. The in vivo relevance of the in vitro findings can then subsequently be verified in a targeted manner.

Two classes of models have been proposed to explain the emergence of asymmetric distribution of PCP proteins on cell boundaries: (i) gradient models, claiming that the asymmetric distribution of PCP proteins reflects the external gradients controlling the level and/or activity of PCP proteins (*Hale et al., 2015*; *Rogulja et al., 2008*), and (ii) localized feedback models, where small initial biases in complex polarity are amplified by a combination of self-enhancing feedback between like complexes and mutual inhibition feedback between opposing complexes (*Amonlirdviman et al., 2005*; *Le Garrec et al., 2006*; *Fischer et al., 2013*; *Mani et al., 2013*; *Burak and Shraiman, 2009*) (*Figure 1A*). Our results strongly support the existence of such localized feedbacks in the Fat4-Ds1 system and suggest a potential mechanism driving these feedbacks.

By analyzing co-culture of cells expressing only Fat4 or Ds1 we can analyze the existence and properties of self-enhancing localized feedbacks in the absence of mutual inhibition. Large scale snapshot analysis (*Figure 2*) and live cell imaging of single pairs (*Figure 3*) show that accumulation of complexes at the boundary exhibits a threshold response to increasing levels of Ds1 - a hallmark of positive feedback. This is the first direct observation of a positive feedback loop in a PCP system. Evidence for the second class of localized feedback, the mutual inhibition between opposing complexes, is provided by our finding that strong polarization is consistently found in cells expressing both Fat4 and Ds1, even in the absence of a strong expression gradient in either of the proteins (*Figure 6*).

What could be the mechanism behind the localized feedbacks on Fat4-Ds1 complexes? The strong polarization and inferred presence of both feedbacks in a minimal synthetic system make it plausible that no other proteins specific to this pathway are involved in the feedback, and favor simple mechanisms that rely on direct interactions between the heterotypic complexes. Such feedbacks

can act either at the level of production or degradation. For example, for the self-enhancing feedback case, complexes can either catalyze production of like complexes, or prevent degradation of like complexes. Using quantitative FRAP analysis we show that unlike unbound Fat4 and Ds1, Fat4-Ds1 complexes are extremely stable, and do not recover or diffuse even after several minutes. This observation suggests that the self-enhancing feedback could at least in part rely on enhanced stability of complexes, possibly through cluster formation. Such enhanced stability of clusters was shown to occur with other cadherin such as E-cadherin (*Zhang et al., 2009*; *de Beco et al., 2009*). Furthermore, FRAP analysis of *Drosophila* Ft/Ds in wing disk junctions also showed enhanced stability on boundary puncta compared to other boundary regions (*Hale et al., 2015*).

Enhanced stability may not be the only mechanism contributing to localized feedbacks. For example, in addition to stabilizing complexes, clusters could also catalyze their formation by acting as a 'diffusion trap', as has also been suggested for E-Cadherin (*Wu et al., 2010*). Moreover, work in *Drosophila* has implicated directional trafficking (*Matis et al., 2014*) and feedbacks through cytoplasmic PCP components (*Amonlirdviman et al., 2005*) as other potential mechanisms for localized feedbacks.

One potential difficulty with the extreme stability of Fat4-Ds1 complexes is that it may be hard for the cells to change their initial polarity, for example to respond to cell divisions or morphogenetic processes. Interestingly, the observed trans-endocytosis of Ds1 into the Fat4 cell (*Figure 1D–F*) may serve as a way to quickly remodel boundaries in spite of their slow turnover by removing large fragments of boundaries with otherwise stable Fat4-Ds1 complexes.

Our finding that polarity of Fat4-Ds1 accumulation at the cell boundary can be observed with optical microscopy through the gap between Fat4 and Ds1 fluorescence (*Figure 5*) is surprising. The expected lengths of the extracellular domains of Fat4 and Ds1 in an extended form are 153 nm and 121 nm, respectively (Fat4 and Ds1 have 34 and 27 cadherin repeats, respectively). Hence, the observed gaps are consistent with the length of Fat4 and Ds1 in an extended conformation. A recent study, however, suggested that Fat4 and Ds1 fold into a compact structure to fit into intercellular gaps (*Tsukasaki et al., 2014*). It is unclear whether these seemingly contradictory observations are due to different methodologies or different cellular contexts. Hence, further experiments are required to determine the structural basis of the observed gaps in our experiments.

We used our synthetic system to show that cells expressing both Fat4 and Ds1 exhibit polarized distribution of these proteins on cell-cell boundaries. Hence, our in vitro setup shows that expression of Fat4 and Ds1 is sufficient to generate polarized boundaries. Unlike in vivo tissues where the direction of polarity is coordinated over extended regions, the direction of polarity of each boundary in our in vitro assay seems to be independent of the polarity of nearby boundaries. It is possible that this difference is due to the relatively large cell-to-cell variability in the expression of Fat4 and Ds1 in cell culture. This variability can lead to local effective gradients of Fat4 and Ds1 biasing the direction of polarity in each boundary. This situation is reminiscent of the disordered organization of denticles in *Drosophila* larvae which was attributed to variability in Ds expression (*Rovira et al., 2015*; *Saavedra et al., 2014*).

Analysis of the direction of Fat4 and Ds1 gradients across each boundary indeed shows that for the unambiguous situation of opposing Fat4 and Ds1 gradients, the polarity (determined independently by the 'rainbow' feature) aligns with the local gradients. For the case where gradients are incompatible (i.e. pointing in the same direction), the polarity may be determined by either the Fat4 or the Ds1 gradients, depending which one is more dominant (e.g. which gradient is stronger).

Our results and conclusions are not limited to the understanding of Fat-Ds signaling but provide a general framework for how complex local interactions between membrane proteins can induce tissue level organization. It remains to be seen whether similar mechanisms are also at play in other systems.

## Materials and methods

### Cloning of Fat4 and Ds1

Human Fat4 and Ds1 cDNA sequence were amplified from the mRNA extracted from MCF7 (ATCC HTB-22, RRID:CVCL_0031) cell line. Several fragments of each gene were amplified and then combined using either restriction enzymes or Gibson assembly (*Gibson et al., 2009*). The citrine and

mCherry were fused to the C-terminal end of the full length Fat4 and Ds1, respectively. Fat4 fusion construct was placed under a CMV constitutive expression promoter, while Ds1 fusion construct was placed under a doxycycline inducible promoter (pcDNA5/TO, T-REx system, Thermofisher).

## Cell culture and transfection

HEK293 (ATCC CRL1753, RRID:CVCL_0063) cells were grown in adherent cultures in Dulbecco's Modified Eagle's Medium (DMEM) supplemented with 10% FBS. MCF7 cells were grown in Eagle's Minimum Essential Medium supplemented with 10% FBS and 0.01 mg/ml human recombinant insulin (Biological Industries). All cells were cultured in a humidified atmosphere of 5% $CO_2$ at 37°C. Stable and transient transfections were performed using *Trans*IT-LT1 reagent (Mirus Bio, Madison, WI) according to the manufacturer's instructions. For Fat4 and Ds1 constructs 1 µg of the plasmid was taken, for the other plasmids the mixture of the desired plasmid (200 ng) with an empty vector (800 ng).

Both HEK293 and MCF7 cells are in the list of commonly misidentified cell lines maintained by the International Cell Line Authentication Committee. These two cell lines serve as a cellular platform for testing the interactions between Fat4 and Ds1 in the current work. These cells were chosen since they are standard epithelial cell lines which are often used in in vitro experiments. Furthermore, HEK293 cells do not endogenously express Fat4 and Ds1 and hence provide an ideal platform to study these proteins.

STR profiling confirmed the authentication of the MCF7 cell line but showed significant genetic modifications in the HEK-Fat4-Citrine and HEK-Ds1-mCherry cells, which is consistent with the known genomic instability of the HEK293 cell line. We note that both cell lines are clonal as they were grown from single cell colonies. The specific properties of these cells are not a contributing factor in our synthetic biology platform, nor do we compare between cell lines. For transparency purposes, we will make these cell lines available through a public repository (e.g. ATCC). All our cell lines tested negative for mycoplasma. Tests were performed using the EZ-PCR mycoplasma kit (Biological Industries, Israel).

## Stable cell line establishment

For the generation of stable cell lines, transiently transfected cells were passaged 24 hr post-transfection in growth medium containing the appropriate selection antibiotics for 10 days (Zeocin (InvivoGen, USA) 100 µg/ml for Fat4-citrine constructs, Blasticidin 5 µg/ml and Hygromicin (AG Scientific, USA) 50 µg/ml for MCF7 and 100 µg/ml for HEK293 cells (AG Scientific,USA) for Ds1-mcherry constructs).

Single cell colonies were generated by limiting dilution in a 96-well plate (2 cells/ml). After a period of two weeks, the plates were screened for positive clones, which were transferred to a new plate for further growing.

A HEK293 cell line expressing both Fat4-citrine and inducible Ds1-mCherry was generated by consecutive transfection and selection processes for both constructs.

## SDS page and western blot

Protein samples were prepared by trypsinization of HEK293 cells ($1 \times 10^6$ cells). The cells were washed by PBS buffer and then lysed by adding 2x Laemmli sample buffer supplemented with 2M urea and boiled for 10 min. Samples of cell extracts were separated by SDS-PAGE (GeBARunner, DNR Bio-Imaging Systems) under reducing conditions using 4–12% gradient polyacrylamide gels (DNR Bio-Imaging Systems) according to manufacturer's instructions.

The samples were then electrophoretically transferred for 16 hr in 4°C using wet transfer standard protocol to the nitrocellulose membranes. Blots were blocked for 1 hr in PBST (PBS buffer and 0.1% Tween) containing 5% skim milk, followed overnight incubation at 4°C with anti-GFP (Cell signaling, RRID:AB_390710) and anti-mCherry (Clontech, RRID:AB_10013483) primary antibodies. Blots were washed three times and incubated for 1 hr at room temperature with the secondary antibody (HRP-conjugated goat anti-rabbit IgG, Jacksonimmuno, RRID:AB_2307391). Immunoreactive bands were visualized by the enhanced chemiluminescence method (ECL) (Biological Industries) according to standard procedures.

## qPCR analysis

HEK-Ds1-mCherry cells were grown for 24 hr. Doxycycline (DOX) was added for 0, 0.5, 1 and 2 hr. RNA was isolated with TRIzol reagent (Life Technologies). 1 μg of RNA was reversed transcribed (SuperScript III, Termo Fisher Scientific). mRNA expression was evaluated with the TaqMan Gene Expression Assay (Applied Biosystems) using the FastStart Universal Probe Master (Roche). Relative expression of the mRNAs was normalized to GAPDH. Real-time PCR was performed in triplicate.

Primers sequence:

For mCherry: Forward: AGGACGGCGAGTTCATCT, Reverse: CCCATFGTC TTCTTCTGCATTA

For GAPDH: Forward: GCTGGCATTGCCCTCAAC, Reverse: CATGAGGTCCAC CACCCTG

## Cell preparation for snapshot analysis and free co-culture experiments, FRAP and FRAP-TIRF experiments

A co-culture/monoculture of Fat4-citrine and Ds1-mCherry expressing cells ($1.6 \times 10^4$ cell/ml) was seeded onto 24-well glass bottom plates (Cellvis, USA) or 35 mm plates (SPL lifesciences, Korea) 12 hr prior the imaging.

For snapshot analysis assay the 24-well plates were covered with 50 ug/ml of Concanavalin A (Sigma Aldrich) to improve cell adherence. Cells were grown for 12 hr and then for induction of Ds1-mCherry, 100 ng/ml doxycycline (Sigma-Aldrich) was added to the growth medium for various periods of time. After that the cells were washed with PBS and fixed for 15 min at room temperature with 2% paraformaldehyde in PBS. To visualize nuclei, the cells were stained with Hoechst Stain solution (Sigma Aldrich) for 5 min.

For FRAP on boundaries and FRAP-TIRF, the cells were seeded on 24-well plates and 35 mm plates, respectively. Directly prior the imaging the media was replaced with low fluorescence imaging media (αMEM without Phenol red, ribonucleosides, deoxyribonucleosides, folic acid, biotin and vitamin B12 (Biological Industries, Israel)). For induction of Ds1-mCherry expression, 100 ng/ml doxycycline (Sigma-Aldrich) was added to the growth medium 12 hr prior to imaging.

## Micropatterns

Micropatterning was performed as previously described (*Shaya et al., 2017*), In brief, A PDMS mold with raised bowtie patterns was attached to a glass surface of the 6-well glass bottom plates (Mat-Tek, USA) after being treated with a UV/Ozone cleaning device (UVOCS, USA). Liquid agarose (0.6% in 2:3 EtOH:ddH$_2$O) was wicked into the gap between the mold and the glass and an inverted pattern of agarose was formed upon removal of the PDMS mold. Bovine Fibronectin (50 μg/ml, Biological Industries) was adsorbed on the exposed regions of the glass by incubating it for 1 hr at room temperature. The square size of the bowties used was $20 \times 20$ μm which yielded the highest probability of a single cell to attach in each half of the bowtie. The mix of HEK293-Fat4-citrine and HEK293-Ds1 cells was diluted to $1.6 \times 10^4$ cell/ml and seeded onto the patterned plate. Directly prior imaging the media was replaced with a previously mentioned low fluorescence imaging media. For induction of Ds1-mCherry, 100 ng/ml doxycycline (Sigma-Aldrich) was added to the growth medium to induce expression.

## Microscopy details

### Imaging of fixed cells for snapshot analysis

Cells were imaged using Nikon TI-E inverted microscope (Nikon, Japan) equipped CFI Plan Apo 20X objective NA = 0.7 (Nikon, Japan); and an ANDOR sCMOS camera (Andor, Belfast, Northern Ireland). The equipment was controlled by Micro-Manager 1.4 software (UCSF). For each field of view 10 planes with 1 μm apart in the z direction were taken.

### FRAP on the accumulating boundaries

FRAP experiments on the accumulating boundaries were performed using Andor revolution spinning disk confocal microscope supplied with FRAPPA device (Andor, Belfast, Northern Ireland). Photo-bleaching was performed with 70% power of the 445 nm laser for a total bleach time of 75 ms (3 repeats of 25 ms).

## FRAP-TIRF experiments

Cells were imaged in FRAP-TIRF iMIC system (Till photonics) equipped with an oil-immersion Plan-Apochromatic 100x objective NA = 1.45 (Olympus, Tokyo, Japan) and an ANDOR iXon DU 888D EMCCD camera (Andor, Belfast, Northern Ireland). FRAP protocol is similar to the one described previously in Khait et al (Khait).

## Live cell imaging of Fat4-Ds1 movies

Cells were imaged using Andor revolution spinning disk confocal microscope (Andor, Belfast, Northern Ireland). The imaging setup consisted of an Olympus inverted microscope with an oil-immersion Plan-Apochromatic 60x objective NA = 1.42 (Olympus, Tokyo, Japan); and an ANDOR iXon Ultra EMCCD camera (Andor, Belfast, Northern Ireland). For Fat4-citrine and Ds1-mCherry co-culture movies (*Figure 3*), 90 images were taken every 10 min with exposure of 500 ms. For each time point 8–12 z planes every 1 μm were taken.

Movies of Ds1-mCherry activation (*Figure 3—figure supplement 2*) were performed using a Zeiss LSM880 confocal microscope. Images were taken every 5 min. For each time point 7 z-planes every 1 μm were taken.

The movie of cells expressing both Fat4-citrine and Ds1-mCherry (*Figure 6—figure supplement 1*) was performed using Zeiss LSM880 confocal microscope equipped with Airyscan detection unit (Carl Zeiss AG, Germany). Image were taken every 5 min.

## High resolution confocal imaging and super resolution imaging

High resolution images were acquired either with a Leica TCS SP5 (Leica Microsystems, Wetzlar, Germany) of with Leice TCS SP8 (Leica Microsystems, Wetzlar, Germany). The SP5 setup included HC PL APO 63x/1.40 Oil STED objective.

For the super resolution images we used Leica TCS SP8 equipped with a STED module and an HC PL APO 100x/1.40 Oil STED objective and white laser. For STED images, the white light laser was set on excitation wavelength of 510 nm or 586 nm with an emission window of 520–581 nm or 605–647 nm, using the 592 nm or 660 nm depletion laser to narrow the PSF of the signal thus improving resolution. Deconvolution of the acquired images was performed with the Huygens Pro software on default settings.

## Data analysis

### Snapshot analysis

Prior to analysis, z stacks of all the images were converted to 'average intensity projection' images. Snapshots of Fat4-Ds1 co-culture were segmented using Ilastik (*Sommer et al., 2011*), an open source software for image classification and segmentation (http://ilastik.org/) and custom written Matlab code. One classifier was trained on the Fat4-citrine and Ds1-mCherry signal to distinguish regions of Fat4 expressing cells, Ds1 expressing cells and background. A second classifier was trained on the co-localization signal of Fat4-citrine and Ds1-mCherry complexes to identify accumulating boundaries with the 'intensity feature' turned off to prevent inadvertently thresholding to identify boundaries and thereby creating a bias. A final classifier identified nuclei based on the Hoechst signal. Cells were then segmented by using the nuclei for a seeded watershed of the Fat4-Ds1 regions. For cells where the classifier failed to identify the cell type, raw Fat and Ds intensity was used to assign the type. Accumulating boundaries were assigned to cell interfaces based on their presence in cells of a dual lattice generated by watershedding the interfaces. Distributions in *Figure 2* were generated from mean intensities in each cell shifted to have the dimmest cells have value 10, a cutoff value used for the log scale binning. Bins were spaced evenly on the log scale so the graphs show P(log I). In 2D histograms a fixed lookup table for the probabilties was used for different time points and intensity distributions for interfaces with and without accumulation were normalized by the sum of the two distributions (the total number of cells). All analysis code can be found at https://github.com/idse/FatDs/ (*Loza, 2017*; a copy is archived at https://github.com/elifesciences-publications/FatDs/). Intensity values used to generate the distributions in *Figure 2* and *figure 2—figure supplement 2* can be found in the alldata.xlsx file.

## Fat4-citrine and Ds1-mCherry accumulation movies

Prior to analysis, z stacks of all the images were converted to 'average intensity projection' images. Fat4-citrine and Ds1-mCherry accumulation movies were analyzed using semi-automatic custom written Matlab code. Cells were first segmented be mapping Fat4-citrine cells (green) and Ds1-mCherry cells (red), then the total fluorescence in these areas was measured. The overlap between the red and green areas was mapped as accumulating boundary. Since Ds1-mCherry fluorescence levels are low at short induction time; the segmentation by the code was not complete. In these cases we manually corrected the segmentation using DIC images for proper cell recognition.

## FRAP analysis of Fat4-Ds1 accumulating boundary

Boundaries were segmented and tracked using the open active contour method (snake) implemented in the ImageJ plugin JFilament (*Smith et al., 2010*) on the unbleached Ds1-mCherry signal. Using custom written Matlab code the Fat4-citrine signal along the snake was obtained as the maximal intensity projection normal to the snake over a nine pixel wide strip. Given the drift and growth/shrinkage of the snakes over times, the signal along the snake at different times had to be registered by maximizing the spatial cross-correlation to produce the space time profile. For Fat4-Ds1 boundaries we bleached half the boundary. We then fit an error function to the profile at each time, adapting (*Goehring et al., 2010*), i.e. at each time we fit:

$$f(x) = U_0 \left(1 - A \frac{1 + \mathrm{erf}\left(\frac{x - x0}{L}\right)}{2}\right)$$

The diffusion constant was then obtained by fitting the linear relation with the squared width of the error function, which follows $L^2 = L_0^2 + 4Dt$, while the exchange rate is the time scale obtained from fitting an exponential decay to the dynamics of A. The reason for doing things this way rather than fitting a spacetime profile directly is an increased robustness against registration errors, growth and shrinkage of the snakes and overall brightness changes caused by boundary drift in z.

## Image analysis of FRAP-TIRF experiments

All data processing was performed using custom written Matlab code, as described in Khait et al. (*Khait et al., 2016*). We used a semi-automatic analysis code for FRAP data extraction and fitting procedure. In brief, we defined for each movie a region of interest around the bleached area. The fluorescence profile as a function of time was extracted, corrected for background level and photobleaching and averaged along the axis parallel to the bleached stripe resulting in 1D fluorescence profile for each time point. Fitting to the recovery profiles for extracting Diffusion coefficients and exchange were done according to Khait et al (*Khait et al., 2016*).

## Rainbow analysis and gradient measurements

As in the boundary FRAP analysis, boundaries for rainbow analysis were segmented using JFilament. Using custom written Matlab code intensities were interpolated on equaly spaced points along the boundary and along lines normal to the boundary to create a straightened intensity image. For each position on the boundary the intensity profile normal to the boundary was then fitted by a Gaussian. The relative position of the Fat and Ds peaks defines the direction of the polarity in *Figure 6*, while the distribution of the distance between the peaks for each position along each boundary is shown in *Figure 5D*. To determine gradients for *Figure 6*, cytoplasmic intensities were determined by taking the summed intensity projection of 5 slices centered round the plane of the rainbow, and then measuring the total intensity in a box away from the boundary, excluding the nucleus using the DAPI image and excluding vesicles using an intensity threshold.

## Acknowledgements

We would like to thank Helen McNeil for advice and support. This work was supported by a grant from the Israeli Science Foundation (grant no. 545/14).

## Additional information

### Funding

| Funder | Grant reference number | Author |
|---|---|---|
| Israel Science Foundation | 545/14 | Olga Loza<br>Nadav Gordon-Bar<br>Liat Amir-Zilberstein<br>Yunmin Jung<br>David Sprinzak |

The funders had no role in study design, data collection and interpretation, or the decision to submit the work for publication.

### Author contributions

Olga Loza, Conceptualization, Data curation, Investigation, Methodology, Writing—original draft, Writing—review and editing; Idse Heemskerk, Conceptualization, Data curation, Investigation, Software, Methodology, Writing—original draft, Writing—review and editing; Nadav Gordon-Bar, Yunmin Jung, Data curation, Investigation; Liat Amir-Zilberstein, Resources, Data curation; David Sprinzak, Conceptualization, Supervision, Investigation, Methodology, Writing—original draft, Writing—review and editing

### Author ORCIDs

Idse Heemskerk, http://orcid.org/0000-0002-8861-7712
David Sprinzak, http://orcid.org/0000-0001-6776-6957

### Decision letter and Author response

Decision letter https://doi.org/10.7554/eLife.24820.021
Author response https://doi.org/10.7554/eLife.24820.022

## Additional files

### Supplementary files

• Transparent reporting form
DOI: https://doi.org/10.7554/eLife.24820.020

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
