## [Decision Letter]

Thank you for submitting your article "A synthetic planar cell polarity system reveals localized feedback on Fat4-Ds1 complexes" for consideration by *eLife*. Your article has been reviewed by two peer reviewers, and the evaluation has been overseen by a Reviewing Editor and Naama Barkai as the Senior Editor. The reviewers have opted to remain anonymous.

The reviewers have discussed the reviews with one another and the Reviewing Editor has drafted this decision to help you prepare a revised submission.

Summary:

The paper reports an original and ambitious effort to reconstitute the Fat/Dachsous polarity system in tissue culture cells. Using fluorescent fusion constructs for these proteins, they manipulate and quantify their total levels over time and quantify their incorporation into polarized complexes at cell contacts. They use these data to quantitatively test models for polarization mechanisms. In particular, the authors address the roles of positive and negative feedback and the role of differential Dachsous expression between cells in organizing polarity of these domains. Their data suggest that 1) The Fat/Ds system spontaneously breaks symmetry at threshold concentrations, 2) heterophilic Dachsous/Fat complexes across cell contacts enhance the formation of complexes of the same polarity and discourage accumulation of oppositely oriented complexes, and 3) that the orientation of these complexes is biased by relative levels of Ds in neighboring cells.

While the reviewers were excited by the potential of this approach, several questions were raised about the data and their interpretation that would need to be addressed before the paper could be considered for publication.

Essential revisions:

1) A key argument for positive feedback is based on evidence in Figure 2 and Figure 3 that Fat4-Ds1 show threshold response binding dynamics. The reviewers raised concerns regarding unexplained anomalies in the data, and think that the data supporting this point need to be strengthened. Specifically….

– How convincing is population level data for threshold effect of Fat4-Ds1 binding? Could Figure 2 equally well fit a single exponential (expected I think for mass action binding?). The authors should better explain how they know that the Hill Coefficient with n=2 is the "best" fit. Also, it seems that the fit shown in Figure 2 represents data from only one experiment. It is important to show that a similar coefficient fits the data of the second experiment.

– In Figure 2, if there is a threshold response for both proteins, shouldn't there be 3 quadrants (bottom left and right, top left) containing purple dotes and one quadrant (top right) containing yellow dots, with the boundaries between the quadrants representing the thresholds? In "No induction" the yellow dots seem scattered evenly almost everywhere, suggesting little or no threshold response in cells where binding is actually seen. But conversely the purple dots suggest there is a second set of boundaries where there is low Fat4 on one side and a wide range of Ds1 levels on the other (which supports a threshold for Fat4 levels but not Ds1 levels?). Look at 20h induction, and now there are lots of cell pairs binding with both high Fat4 and high Ds1, but strangely no cell pairs not binding with low Ds1 and high Fat4 (even though low Ds1 cells exist) – this cell category is just missing which seems to require some explanation. Does the fact that there do not seem to be any points (purple or yellow) where Fat is high but Dachsous is low mean that Fat levels are indirectly increased (perhaps by stabilization) by the growing amounts of Dachsous in the other pool of cells. These data need to be better explained an accounted for.

– The experiments shown in Figure 3 and Figure 3—figure supplement 1 and 2 are meant to show using live imaging that complex formation begins at a threshold expression level. A puzzling aspect of the experiment in Figure 3 is that boundary accumulation seems to begin before there is a detectable rise in Ds levels. A related experiment shown in the Figure 3 supplement does not present the fat and ds intensity data from the start of Ds induction but only from two hours after induction so we can't see at what Ds threshold the complexes begin to accumulate.

A problem with interpreting this type of experiment is that the total fluorescence intensity in the cell likely reflects pools of ds in the secretory and endocytic pathways. So we don't really know the amount of ds that is actually at the surface and capable of interacting with fat. This is the pool that is relevant to the authors' model. A key piece of data necessary to interpret this experiment is some measurement of the surface level of ds (in the absence of fat expressing cells) at different times after induction.

The experiment in Figure 6 nicely shows that polarized fat/ds complexes tend to have the same polarity on the same cell boundaries and that the ds expression difference between neighboring cells strongly influences the orientation of polarized fat/ds complexes. The authors argue from this that the difference in ds expression can orient polarity and that there is mutual inhibition between complexes with different polarities. To strongly conclude the latter, it would be necessary to monitor the formation of these polarized domains in timelapse.

The authors suggest that transendocytosis of Fat/Ds complexes occurs based on an image shown in Figure 1. They do not present evidence that these spots of fluorescence are within endosomes rather than outside cells. This should be shown, or the data could be removed as it is not essential for the conclusions of the paper.

---

## [Author Response]

*Essential revisions:*

*1) A key argument for positive feedback is based on evidence in Figure 2 and Figure 3 that Fat4-Ds1 show threshold response binding dynamics. The reviewers raised concerns regarding unexplained anomalies in the data, and think that the data supporting this point need to be strengthened. Specifically….*

*– How convincing is population level data for threshold effect of Fat4-Ds1 binding? Could Figure 2 equally well fit a single exponential (expected I think for mass action binding?). The authors should better explain how they know that the Hill Coefficient with n=2 is the "best" fit. Also, it seems that the fit shown in Figure 2 represents data from only one experiment. It is important to show that a similar coefficient fits the data of the second experiment.*

The fit is a standard fit (least square method in Matlab) with two parameters, k and n. The best fit Hill coefficient, n, is automatically determined with 95% confidence interval providing a measure of the likelihood that the "real" function has a Hill coefficient in this range. The 95% confidence interval for the first experiment is [1.7 2.3] meaning that the change the real value is less than 1.7 is 2.5%. As mentioned by the reviewer the expected Hill coefficient for mass action kinetics is n=1 (Michaelis Menten) so this value is very unlikely given the data. Using a different measure, running a fit with fixed n=1 results in a significantly lower ‘goodness of fit’ characterized by the sum of square errors (sse). The sse for the current fit is 0.076 and the sse for n=1 fit is 0.144.

Fitting the second data set, which exhibited lower Ds induction (due to experimental variability) resulted in a higher Hill coefficient of n=4.5±1.3 (new Figure 2—figure supplement 2). While the result is not the same, the Hill coefficient in the second set is also significantly higher than 1.

We note that population variability would smooth out any threshold, so that these Hill coefficients serve as a lower bound: the true value will almost certainly be higher.

*– In Figure 2, if there is a threshold response for both proteins, shouldn't there be 3 quadrants (bottom left and right, top left) containing purple dotes and one quadrant (top right) containing yellow dots, with the boundaries between the quadrants representing the thresholds? In "No induction" the yellow dots seem scattered evenly almost everywhere, suggesting little or no threshold response in cells where binding is actually seen. But conversely the purple dots suggest there is a second set of boundaries where there is low Fat4 on one side and a wide range of Ds1 levels on the other (which supports a threshold for Fat4 levels but not Ds1 levels?). Look at 20h induction, and now there are lots of cell pairs binding with both high Fat4 and high Ds1, but strangely no cell pairs not binding with low Ds1 and high Fat4 (even though low Ds1 cells exist) – this cell category is just missing which seems to require some explanation. Does the fact that there do not seem to be any points (purple or yellow) where Fat is high but Dachsous is low mean that Fat levels are indirectly increased (perhaps by stabilization) by the growing amounts of Dachsous in the other pool of cells. These data need to be better explained an accounted for.*

We thank the reviewers for pointing out the inconsistency in our data. We have carefully checked the analysis and found a problem (bug in the code) with the 2D distribution plot of the non-accumulating boundaries (purple dots). We have also re-examined the binning and background subtraction procedures and reanalyzed the whole dataset with more well-defined procedures (now described in the Materials and methods).

The new analysis shown in Figure 2 and Figure 2—figure supplement 1 shows that non-accumulating boundaries spread over all quadrants almost symmetrically. It also shows that the Ds distribution spreads to higher Ds levels as Ds is induced.

Analysis of the second dataset (Figure 2—figure supplement 2) is now provided. We note that in this dataset the lower right quadrant has fewer boundaries. Although we are not certain why this is the case, we suspect that this probably due to a somewhat lower cell density in this experiment allowing cells to move around potential bind to high expressing partners. Despite the difference between the two sets, the main points are still valid: We see a non-linear response of the fraction of boundaries (Figure 2—figure supplement 2) and the accumulating boundaries require both high levels of Fat4 and Ds1 (Figure 2—figure supplement 2). Hence, both datasets support threshold response.

The reason the yellow dots were spread out in the no induction case was because there were so few accumulating boundaries it was basically showing noise, but the distribution was normalized by its own total making the noise look bright. By normalizing the yellow and purple distributions not by their own total, but their joint total, the intensity of the yellow distribution now represents its mass relative to the total number of Fat-Ds interfaces, which means the intensity now represents actual fraction of accumulating boundaries and does not amplify noise when there is no accumulation.

Finally, we checked whether there is a stabilization of Fat4 in accumulating boundaries by looking at the mean level of Fat4 as a function of time. We see only a small increase, about 10%, in total Fat4 levels as Ds1 is induced. Hence, formation of boundaries do not significantly affect the total lifetime of Fat4 in our system.

*– The experiments shown in Figure 3 and Figure 3—figure supplement 1 and 2 are meant to show using live imaging that complex formation begins at a threshold expression level. A puzzling aspect of the experiment in Figure 3 is that boundary accumulation seems to begin before there is a detectable rise in Ds levels. A related experiment shown in the Figure 3 supplement does not present the fat and ds intensity data from the start of Ds induction but only from two hours after induction so we can't see at what Ds threshold the complexes begin to accumulate.*

We have now investigated carefully the turn-on point in Figure 3 and Figure 3—figure supplement 1. Indeed, the Observed Ds level starts increasing above background about 100min after Dox induction (see new Figure 3—figure supplement 2). The reason for this is probably the finite maturation time of mCherry and the time it takes for the accumulating Ds1 to reach threshold detection levels (We have checked that expression level increases linearly as Dox is induced – see qPCR results in supplement 2D).

Although in some movies it seems that accumulation on the interface occurs as Ds1 starts to increase, the real signature feature of positive feedback at the threshold is the fact that accumulation on the interface is not proportional to the total rate of increase but instead much faster.To make this point we now plot Ds1 accumulation as a function of total Ds1 level for the three movies shown in the figure (Figure 3—figure supplement 1). We indeed see that the slope in log-log plot, is significantly higher than 1 (dashed line) for several hours after induction is observed (much longer than the observed delay in Ds1 induction). For linear mass action kinetics (without feedback) it is expected that the slope in the graph equal to 1. Having a higher slope means sharper threshold response, or higher Hill effective Hill coefficient. We now address this point in the text.

Regarding the measurement starting at a time higher than 0. The time indicates the time from adding doxycycline. The reason it does not start at 0 is that these movies were taken on free co-cultures, where it is very hard to identify and track potential interacting pairs over long times. Hence, since in the first 2 hours it is less likely to capture accumulation, we started these movies at this time. We note though that these movies do capture the accumulation from the first time it appears. We also note that the description in the captions of both the main figure (were the movie started right after addition of dox) and the supplementary figures was unclear. We have now fixed the captions to make it clear.

*A problem with interpreting this type of experiment is that the total fluorescence intensity in the cell likely reflects pools of ds in the secretory and endocytic pathways. So we don't really know the amount of ds that is actually at the surface and capable of interacting with fat. This is the pool that is relevant to the authors' model. A key piece of data necessary to interpret this experiment is some measurement of the surface level of ds (in the absence of fat expressing cells) at different times after induction.*

We thank the reviewers for the point raised here. We have now performed confocal movies of Ds1 induction at different z-planes, in the absence of Fat4 cells and analyzed the membrane Ds1 levels vs the total Ds1 levels in the cells (new Figure 3—figure supplement 2). The movies clearly show that a large fraction of Ds1 is present at the membrane. This is most clearly seen when looking at the lowest z-plane showing fluorescence from the basal membrane (new Figure 3—figure supplement 2 – see z=0 planes). Furthermore we show (Figure 3—figure supplement 2) that the mean fluorescence from the lowest z-plane (membrane Ds1 level) is proportional to the mean fluorescence from all the z-plane (Membrane + Cell Ds1 level).

As mentioned above, both the membrane and the whole cell Ds1 fluorescence start increasing at about 100min after induction. This is probably due to the maturation time of Ds1 and the time it takes to reach a levels of expression above background.

*The experiment in Figure 6 nicely shows that polarized fat/ds complexes tend to have the same polarity on the same cell boundaries and that the ds expression difference between neighboring cells strongly influences the orientation of polarized fat/ds complexes. The authors argue from this that the difference in ds expression can orient polarity and that there is mutual inhibition between complexes with different polarities. To strongly conclude the latter, it would be necessary to monitor the formation of these polarized domains in timelapse.*

We thank the reviewers for this comment. Clearly the dynamics of rainbow formation is an exciting topic which we plan to address in the future. It is technically challenging to dynamically follow the rainbow features, as there is a tradeoff between spatial resolution and time resolution (i.e. high quality images take time). However, we have managed to get as an example a nice movie showing the turning on of the accumulation and the emergence of rainbow (This movie was taken with Airyscan technology). As can be seen in Figure 6—figure supplement 1, the boundary turns on and become polarized within a couple of hours. Consistent with the data in fixed samples that polarity of the rainbow aligns with the Ds1 gradient. We note that, as expected, the differences in Ds1 expression existed prior to the formation of the rainbow. We now discuss this point in the text.

In general, we agree that understanding the dynamics of polarity is interesting, but we feel that it is beyond the scope of the current manuscript.

*The authors suggest that transendocytosis of Fat/Ds complexes occurs based on an image shown in Figure 1. They do not present evidence that these spots of fluorescence are within endosomes rather than outside cells. This should be shown, or the data could be removed as it is not essential for the conclusions of the paper.*

Transendocytosed vesicles all appear within the cell. The image in Figure 1 is taken from a middle z-plane. Furthermore, co-culture movies as the one shown in Figure 3 and Figure 4, all show that these trans-endocytosed vesicles are moving inside the cell (see yellow vesicles Fat4 expressing cell) and exhibit similar dynamics as Ds1 containing vesicles in the cells. All this supports that these are indeed real intracellular events and not artifacts.

While we agree that this is not the main focus of the current paper, we still think it is important to report this observation, as it is the first observation of transendocytosis in this system.